# Cost-effectiveness of decellularised bone allograft compared with fresh-frozen bone allograft for acetabular impaction bone grafting during a revision hip arthroplasty in the UK

Kern Cowell ,[1] Patrick Statham,[1] Gurdeep Singh Sagoo ,[2,3] James H Chandler,[4] Anthony Herbert,[1] Paul Rooney,[5] Ruth K Wilcox,[1] Hazel L Fermor[6]

For numbered affiliations see end of article.

**Correspondence to**
Dr Hazel L Fermor;
h.l.fermor@leeds.ac.uk

## ABSTRACT

**Objectives** Fresh-frozen allograft is the gold-standard bone graft material used during revision hip arthroplasty. However, new technology has been developed to manufacture decellularised bone with potentially better graft incorporation. As these grafts cost more to manufacture, the aim of this cost-effectiveness study was to estimate whether the potential health benefit of decellularised bone allograft outweighs their increased cost.

**Study design** A Markov model was constructed to estimate the costs and the quality-adjusted life years of impaction bone grafting during a revision hip arthroplasty.

**Setting** This study took the perspective of the National Health Service in the UK.

**Participants** The Markov model includes patients undergoing a revision hip arthroplasty in the UK.

**Intervention** Impaction bone grafting during a revision hip arthroplasty using either decellularised bone allograft or fresh-frozen allograft.

**Measures** Outcome measures included: total costs and quality-adjusted life years of both interventions over the lifetime of the model; and incremental cost-effectiveness ratios for both graft types, using base case parameters, univariate sensitivity analysis and probabilistic analysis.

**Results** The incremental cost-effectiveness ratio for the base case model was found to be £270 059 per quality-adjusted life year. Univariate sensitivity analysis found that changing the discount rate, the decellularised bone graft cost, age of the patient cohort and the revision rate all had a significant effect on the incremental cost-effectiveness ratio.

**Conclusions** As there are no clinical studies of impaction bone grafting using a decellularised bone allograft, there is a high level of uncertainty around the costs of producing a decellularised bone allograft and the potential health benefits. However, if a decellularised bone graft was manufactured for £2887 and lowered the re-revision rate to less than 64 cases per year per 10 000 revision patients, then it would most likely be cost-effective compared with fresh-frozen allograft.

## STRENGTHS AND LIMITATIONS OF THIS STUDY

⇒ By using the widely used Markov model, this study captured the dynamic nature of revision hip arthroplasty outcomes and estimated the costs and quality-adjusted life years over the lifetime of the patient.

⇒ The study included sensitivity analyses that assessed the variability and uncertainty of the findings by exploring the impact of key variables on the cost-effectiveness of the grafts used.

⇒ The use of the stochastic modelling with 10 000 simulations provides a strong evaluation of the estimates made by accounting for parameter variability.

⇒ One potential limitation is the lack of clinical data meaning the study relies on assumptions and extrapolation from studies involving different graft types for the decellularised graft revision rates.

⇒ Another potential limitation is the use of costs from a variety of sources which may not accurately reflect the costs paid by the National Health Service in 2022.

## INTRODUCTION

In the UK, there are approximately 8000 revision hip arthroplasties (RHAs) each year;[1] 28% of RHAs require an impaction bone graft (IBG) to replace removed bone before implantation of the joint components.[2] The current gold-standard bone graft is an autograft.[3] However, as the average amount of bone tissue needed for an IBG is over two femoral heads per revision, allograft bone is most often used.[2] As allograft bone contains donor cellular material, there is a risk of a host immune reaction to the graft, which reduces regeneration, revascularisation and incorporation of the graft.[4] In the UK, 10% of RHAs require further revision,[1] with the most common reason being aseptic loosening

(39% of cases).[5] The use of an IBG during an RHA further increases the risk of aseptic loosening due to poor graft incorporation, infection and acetabular cup loosening.[6] If acetabular graft incorporation could be improved, then there is potential to increase the success of RHA and reduce the number of further revisions.[7]

Decellularisation is a novel technology that removes cells and DNA from tissue while preserving the underlying tissue matrix.[8] Depending on the manufacturing method used, decellularised bone allografts can retain the advantageous biological aspects of allografts such as osteoconductive growth factors and have similar mechanical properties to a fresh-frozen allograft, without evoking an adverse immune response.[9] These properties suggest that a decellularised IBG could have increased rates of regeneration and incorporation with the host environment, potentially decreasing the occurrence of aseptic loosening of hip replacement components.[10]

However, the overall costs of an RHA with decellularised allograft are £39 017, compared with the overall cost of fresh-frozen allograft IBG: £16 343, there is an increase of £22 674 per surgery. Therefore, any potential health benefit from decellularisation needs to be evaluated against the potential increase in cost. In this study, a cost-effectiveness analysis was conducted to compare the novel decellularised allograft with the current most common bone graft option (fresh-frozen allograft) for acetabular IBG during an RHA. The costs and health benefits of both bone graft choices were estimated and compared using a Markov model from the perspective of the UK National Health Service (NHS).[11]

## METHODS
### Economic modelling
A decision tree and Markov model[11] was constructed to estimate the costs and benefits of both graft choices (figure 1). The model has yearly cycles that estimate the quality-adjusted life years (QALYs) and costs over the lifetime of the patient from their first RHA shown by the decision tree. At year 0 of the model, all patients start at the beginning of the decision tree at their first RHA; in the following years, the patients transition between the 're-revision' and 'post-revision' health states or enter the 'death' health state. A discount rate of 3.5% per year for both cost and effectiveness was used in line with the recommendations from the National Institute for Health and Care Excellence (NICE) guide for technology appraisal.[12] The time spent in each health state was calculated and the weighted costs and QALYs of each health state were totalled to give the overall cost and QALYs for each bone graft choice.

To compare the two bone graft choices, an incremental cost-effectiveness ratio (ICER) was used. The ICER is calculated by:

$$ICER = \frac{C_1 - C_0}{E_1 - E_0} \qquad (1)$$

where $C_1$ is the cost and $E_1$ is the effectiveness of the new intervention with $C_0$ and $E_0$ being the cost and effectiveness of the original intervention.[13] In the UK, new treatment adoption decisions are guided by NICE through a technology appraisal which considers the ICER. If the ICER is below £20 000, the new treatment is

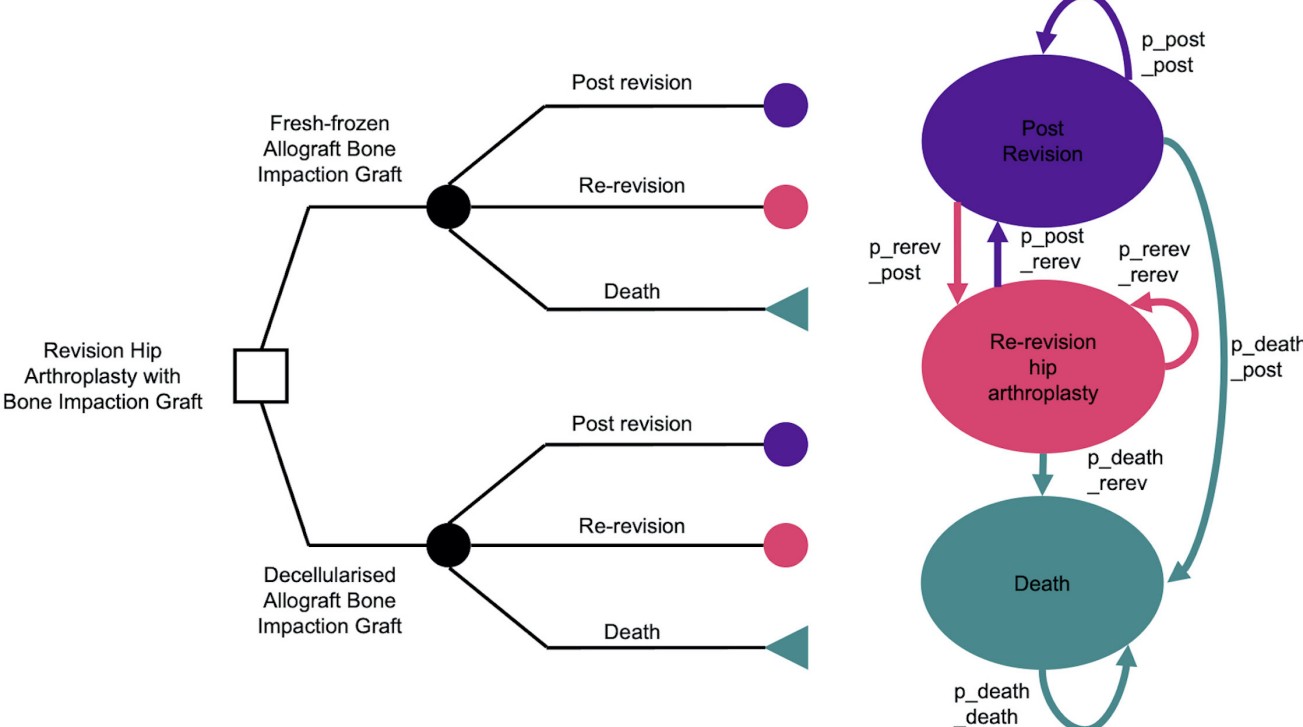

**Figure 1** Decision tree and Markov model of post-revision hip arthroplasty health states: triangle end states show where decision tree enters Markov model; transition probabilities labelled as p_(to health state)_(from health state).

**Table 1** Price of decellularised femoral head allograft: all costs are in GBP (pound sterling) and converted into 2022 prices,[25] selling price based on average medical technology operating profit of 19.4%[26]

| | | Cost per graft | Sources |
|---|---|---|---|
| Product cost | Tissue | 709.71 | 14 |
| | Reagents | 557.56 | Online supplemental table 2 |
| | Product labour | 12.12 | 27 |
| | Lab rent | 219.75 | 28 29 |
| | Consumables and packaging | 30.48 | |
| | Amortised investment | 7.18 | |
| Indirect overheads | | 1536.80 | |
| Sales and distribution | | 5020.36 | |
| Selling price | | 10040.72 | |

considered cost-effective, and its adoption is often recommended. An ICER of £20 000–30 000 indicates the need for more certainty around the estimated ICER or added benefits beyond the recorded QALYs, for example., life-extending treatment at the end of life. An ICER above £30 000 requires a substantial reason for the technology to be adopted, for example, it is an innovative technology that has benefits beyond the currently evaluated health benefits.[12]

## Costs

To find the cost of each bone graft choice, the cost of an RHA was combined with the cost of each bone graft (table 1 and online supplemental table 1). The individual costs of purchasing each graft were calculated for one femoral head, with the cost of 2.43 femoral head grafts used for each revision surgery (the average number used per procedure).[2] As the NHS provides free public healthcare to residents of the UK, all estimated costs used in this study are those commonly paid by the NHS, including all inpatient stay costs, aids and adaptations and medication. The source used for the surgical costs did not include surgeon fees and therefore they were not included in this study.[14] Due to the same surgical time being used for both grafts, including surgeon fees would not change the results of the model.

The costs of the reagents used for the decellularisation process were obtained from a variety of suppliers (online supplemental table 2). The costs of the first revision surgery and the 're-revision' health state were estimated under the assumption that the same bone impaction technique was used for each bone graft choice. The 'post-revision' health state has a cost of £54.19 per patient per year. This is representative of the average cost of rehabilitation, medication and other health service costs during the years after an RHA.[15]

## Utility

For this analysis, QALYs are used as the indicator of effectiveness and health benefit. QALYs incorporate the impact of a disease and treatment on both the quantity and quality of life and are the recommended outcome measure in the UK.[12] QALY values for the 're-revision' and 'post-revision' health states were calculated from the preoperative ('re-revision') and postoperative ('post-revision') patient-reported outcome measures for revision total hip replacement.[16] Patients completed the EQ-5D-3L, a generic health-related quality of life measure, 6 months to a year after their RHA. As the patient-reported outcome measures data only give an overview of all revision hip replacement patients before and after their surgery, it was assumed in the base case analysis that the QALYs for the 're-revision' (0.397) and 'post-revision' (0.685) health states for both bone graft choices were the same.

## Transition probabilities

The transition probability is the probability that a patient will move from one health state to another during each period of the model (1 year). Online supplemental table 3 shows the transition probabilities used for both bone graft choices in the model.

It was assumed that the risk of needing an RHA is the same for all health states for each bone graft choice. For the fresh-frozen allograft, the risk of re-revision was calculated from the studies summarised in online supplemental table 4 (clinical studies that include acetabular IBG with a minimum follow-up period of 10 years). The transition rate ($T_R$) and transition probabilities ($T_P$) were calculated by:

$$T_R = \frac{-\ln\left(1 - Revision\ Rate\ (\%)\right)}{Follow-up\ time\ (years)} \quad T_P = 1 - e^{-T_R} \quad (2)$$

To date, there have not yet been any clinical studies using decellularised bone as an IBG during RHA.[10] Therefore, to calculate the transition probabilities of re-revision, studies that tested demineralised bone in combination with allograft bone that had a minimum follow-up period of 10 years were chosen. These criteria were chosen as adding demineralised bone matrix to allograft bone reduces the stiffness of the graft but increases the concentration of growth factors. For the purposes of this study, it was postulated that properties of demineralised bone matrix in combination with allograft bone would be similar to the properties of a decellularised bone graft. Only one study with this criterion was found in the literature, Hamadouche et al[17] recorded an 8% revision rate for an IBG of demineralised bone matrix with allograft in 60 RHA patients after a 13-year follow-up. This gave a transition probability of 0.0064 for the 're-revision' health state. Another study with a follow-up of 2.25 years was also found;[18] however, the majority of re-revisions occurred in the first 2 years.[5] Therefore, this study was not included in this cost-effectiveness analysis as a short follow-up period

can significantly overestimate the calculated transition probability.

The transition from the health states to the 'death' state is based on the mean age of the patients in the model. The starting age used for the model was 71 years, this is the average age of RHA patients in the UK National Joint Registry.[1] A national life table was used to estimate the yearly transition probability for each year of the model.[19] As the base case model had an average starting age of 71.7 years, the transition probability from the 'post-revision' health state to the 'death' state started at 0.0195 for the first year and rose to 0.364 for year 29 of the model (subsequent years above patient age of 100 had a transition probability of 1). Due to the surgical risk and hospital care associated with an RHA, the transition probability for the first revision and 're-revision' health states to the death state was different for the first 90 days from the date of surgery, this changed the yearly transition probability to 0.047 for year 1 to 0.306 for year 29 of the model.[20] The combination of transition probabilities from each health state must equal 1. Therefore, the 'post-revision' transition probabilities for each year were calculated by subtracting the probability of entering the 're-revision' and 'death' health states from 1.

### Univariate sensitivity analyses

To test which of the parameters affect the results of the model, a univariate sensitivity analysis was used. One variable in the model was changed to an extreme value and a new ICER was calculated. As suggested in the NICE guide for technology appraisal,[12] the discount rate can significantly affect long-term economic models and diminishes long-term benefits. Therefore, a sensitivity analysis with a lower discount rate of 1.5% was completed. Other parameters of substantial uncertainty that were tested using sensitivity analysis were: the age of patients receiving an RHA; yearly post-surgery care costs and the rate of re-revision of the decellularised graft.

### Stochastic model

Many of the parameters used in the Markov model have varying levels of uncertainty; therefore, distributions of potential values were found using the mean and SD of each parameter (online supplemental table 5). To test the effect of this uncertainty on the ICER, 10 000 models were run using randomly generated values within each parameter distribution.

### Patient and public involvement

Patients or the public were not involved in the design, or conduct, or reporting, or dissemination plans of our research.

## RESULTS
### Cost-effectiveness analysis

For the base case parameters in this model, the overall revision surgery costs for the decellularised allograft were £39 017 per patient and the costs for the fresh-frozen allograft were £16 343 per patient. The QALYs calculated for the decellularised allograft choice were 6.93 per patient with the fresh-frozen allograft providing 6.86 QALYs per patient. To determine the cost-effectiveness of treatment options, the total costs and health benefits of each graft choice model are compared. For the base values in this model, neither bone graft option was found to be dominant, with the decellularised allograft costing more but providing an increase in QALYs (equivalent to 28 days of perfect health over their lifetime). For the base model, the ICER of decellularised allograft was calculated as £270 059 per QALY, which is £240 059 higher than the NICE £30 000 per QALY guidance.[12]

**Table 2** Univariate sensitivity analysis: the incremental costs, QALYs and subsequent ICER from changing one parameter in the model to an extreme value

| Scenario | Incremental costs | Incremental QALYs | ICER (cost per QALY) | Favoured intervention |
|---|---|---|---|---|
| Base case | £20 834 | 0.077 | £270 059 | Fresh-frozen |
| Discount rate changed from 3.5% to 1.5% | £20 511 | 0.092 | £223 491 | Fresh-frozen |
| Low patient cohort age (mean–SD): 60.54 years | £20 110 | 0.133 | £151 109 | Fresh-frozen |
| High patient cohort age (mean+SD): 82.86 | £21 584 | 0.035 | £623 920 | Fresh-frozen |
| Low post-revision care costs: £0 per year | £20 821 | 0.077 | £269 896 | Fresh-frozen |
| High post-revision care costs: £301 per year | £20 895 | 0.077 | £270 859 | Fresh-frozen |
| Same re-revision rate | £28 783 | 0.000 | – | Fresh-frozen |

ICER, incremental cost-effectiveness ratio; QALYs, quality-adjusted life years.

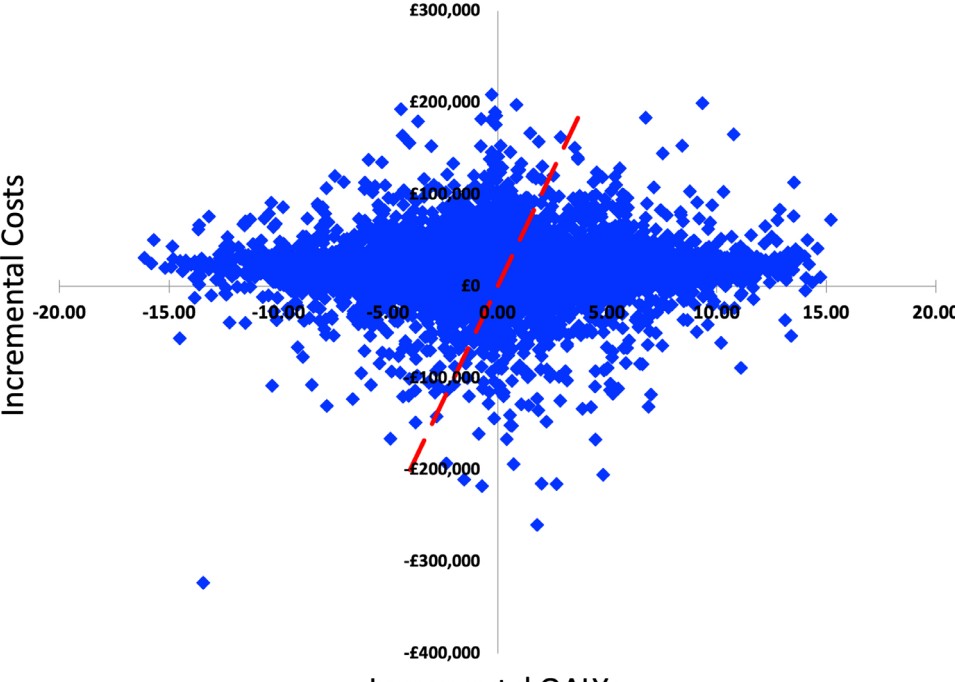

**Figure 2** Stochastic model estimates: incremental costs and QALYs for each of the 10 000 randomly generated parameter models. QALYs, quality-adjusted life years.

## Univariate sensitivity analyses

As can be seen in table 2, the primary parameters that affected the ICER during the sensitivity analysis were the discount rate, average age of the patient cohort and the revision rate. On the other hand, the cost of care during the 'post-revision' health state had little effect on the ICER.

## Stochastic model

Calculated from the results of the randomly generated models, the percentage of models in which the decellularised graft was cheaper than fresh-frozen was 11.1% and the percentage in which the decellularised graft was more effective was 51.2%; this is highlighted in figure 2 by the number of models below the x-axis (cheaper) and to the right of the y-axis (more effective). The average incremental cost and effectiveness of the models was found to be £20 620 and 0.089 QALYs. This gave an ICER estimate of £230 790 per QALY, £39 269 less per QALY than the initial base case model. The cost-effectiveness acceptability curves in figure 3 estimate the probability that one treatment is more cost-effective than the other for different ICER values. The probability that the decellularised graft was cost-effective at the £30 000 per QALY threshold was 41.5% compared with 58.5% for fresh-frozen being cost-effective.

## DISCUSSION

The aim of this study was to evaluate the cost-effectiveness of using a decellularised bone allograft for IBG during an RHA from the perspective of the NHS. For the NICE £30 000 per QALY guidance, both base case and stochastic modelling indicated that fresh-frozen allograft is more likely to be cost-effective than the decellularised graft. However, although the favoured bone graft choice stayed the same throughout the univariate sensitivity analysis, there were large changes to the ICER when the patient cohort age was changed, indicating how substantial any potential uncertainty around the input parameter values could be. However, the stochastic model should effectively capture parameter uncertainty by collating the results of numerous simulations, enhancing the reliability of cost-effectiveness estimates.

Logically, if the cost of producing a decellularised graft was reduced, this would increase the probability of the graft being cost-effective. The current model assumes a large cost of retrieval of the femoral head bone graft (£710 per graft[14]). If an already functioning human tissue provider was to adopt decellularisation of their bone graft products, then this cost could be significantly reduced. The current cost estimate is based on a manual laboratory-based decellularisation system, a commercial operation to manufacture decellularised bone grafts could benefit from cost savings using a large-scale auto-mated manufacturing system. Producing decellularised grafts of tibial and femoral condyle bone for alternative applications alongside femoral head bone grafts would further reduce the costs of a decellularised femoral head graft, with costs shared across different grafts. Similarly, the cost per graft could be further reduced if other tissues were concurrently decellularised using the same manufacturing system. An overhead cost that was also not included in this study is the cost of storing the bone grafts. As decellularised bone grafts are sterilised during

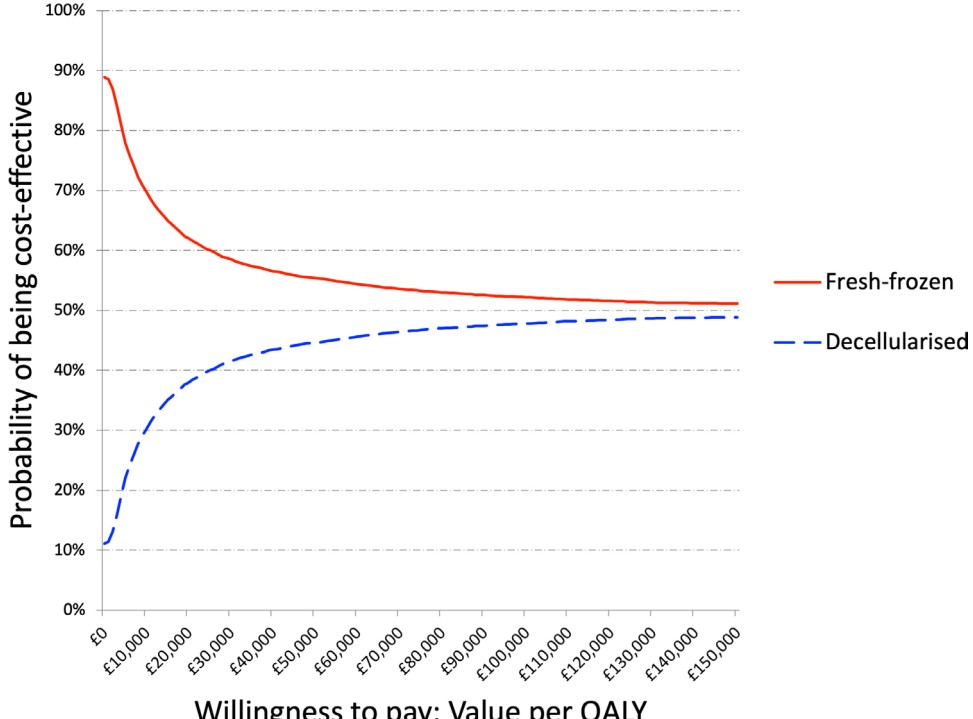

**Figure 3** Cost-effectiveness acceptability curves: the probability that each bone graft choice is cost-effective at varying willingness to pay thresholds. QALY, quality-adjusted life year.

manufacturing, the grafts can be stored at room temperature, whereas fresh-frozen allografts require storage at −80°C. Additionally, once decellularised, human tissue grafts are not considered to be relevant material under the Human Tissue Act,[21] and therefore do not require the additional costs of licensing and tracking management that fresh-frozen allografts require.

Like the uncertainty around the costing of the decellularised graft, the effectiveness of the decellularised graft has also been estimated using many assumptions. As there are currently no clinical studies evaluating the outcome of a decellularised bone graft for IBG during an RHA,[10] the probability of requiring a further revision was estimated from a study of demineralised bone matrix with allograft bone.[17] However, decellularised bone grafts have better mechanical properties than demineralised bone matrix and allograft bone for high load applications and have reduced immunogenicity.[22] Therefore, the model used in this cost-effectiveness analysis could have underestimated the health benefit of the decellularised bone graft.

Additionally, this study's utility measurement relies on QALYs estimated from generic RHA patient data and does not indicate the specific health utility of IBG patients for either graft type. While this simplification was necessary due to a lack of direct clinical data, this could influence the results of the economic model. Furthermore, the study's cost assumptions are based on various sources from across different years, while the costs have been converted to 2022 GBP (pound sterling) prices, these assumptions may not accurately reflect the 2022 costs spent by the NHS. Moreover, the use of the good outcome care costs from Arden et al[19] for the post-revision

group in the model assumes that any potential bad outcome of the RHA would require a re-revision, whereas a poor outcome could be possible that does not warrant revision but has more care costs. However, the sensitivity analysis (table 2) shows any potential biases from this assumption have been shown to have a minor effect on cost-effectiveness.

An alternative to decellularised allograft is washed bone. Studies have found that allograft bone that undergoes a washing process to remove bone marrow possesses similar mechanical properties to fresh-frozen allografts and the removal of immunogenic material could minimise immune reaction and increase bone regeneration.[23] Washed bone products have been provided by NHS Blood and Transplant, Tissue and Eye Services and used by orthopaedic surgeons since 2000.[24] As there have been no reports of negative outcomes with these products, it can be assumed that they perform well, and their continued use suggests that washed bone has clinical benefits over traditional fresh-frozen allograft bone. The cost-effectiveness of washed bone has yet to be systematically evaluated and compared with fresh-frozen or decellularised allograft bone.

### CONCLUSION

The results of this cost-effectiveness analysis suggest that using the current manual manufacturing method, decellularised bone grafts are unlikely to be cost-effective compared with the current fresh-frozen allograft choice. However, if the purchase price of a decellularised bone allograft was below £2887 (71.2%

reduction of the current price) and the graft adequately lowered the re-revision rate of hip arthroplasty to 64 re-revisions per year per 10 000 patients as estimated, the ICER would be below the lower NICE guideline of £30 000 per QALY and the graft likely to be cost-effective. In conclusion, the model-based economic evaluation undertaken in this study suggests that if the cost of manufacturing decellularised bone allograft could be reduced, this would warrant exploration into the effectiveness of these grafts through a randomised clinical trial of IBG during RHA.

**Author affiliations**
¹Institute of Medical and Biological Engineering, Faculty of Engineering and Physical Sciences, University of Leeds, Leeds, UK
²Academic Unit of Health Economics, University of Leeds, Leeds, UK
³Population Health Sciences Institute, Faculty of Medical Sciences, Newcastle University, Newcastle upon Tyne, UK
⁴Institute of Design, Robotics and Optimisation, Faculty of Engineering and Physical Sciences, University of Leeds, Leeds, UK
⁵Research and Development, NHS Blood and Transplant Tissue and Eye Services, Speke, UK
⁶Institute of Medical and Biological Enineering, School of Biomedical Sciences, University of Leeds, Leeds, UK

**Contributors** KC, GSS, PS, JHC, AH, PR, RKW and HLF contributed to the conception. KC, GSS, PS, JHC, AH, PR, RKW and HLF contributed to the study design and methodology. KC and PS contributed to the acquisition. KC, GSS, PS and HLF contributed to the analysis. KC, GSS, PS and HLF contributed to the interpretation of data. KC drafted the work. KC, GSS, PS, JHC, AH, PR, RKW and HLF critically revised the work. KC, GSS, PS, JHC, AH, PR, RKW and HLF approved the final version to be published and agree to be accountable for all aspects of the work. HLF is the guarantor for the project.

**Funding** This study is funded by the Engineering and Physical Sciences Research Council (no. EP/L014823/1).

**Competing interests** None declared.

**Patient and public involvement** Patients and/or the public were not involved in the design, or conduct, or reporting, or dissemination plans of this research.

**Patient consent for publication** Not required.

**Ethics approval** This was a modelling study and ethical approval from the institutional review board was not obtained because no real human participants or animal subjects were involved.

**Provenance and peer review** Not commissioned; externally peer reviewed.

**Data availability statement** Data are available upon reasonable request.

**ORCID iDs**
Kern Cowell http://orcid.org/0000-0003-1737-0836
Gurdeep Singh Sagoo http://orcid.org/0000-0003-1427-1437

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
