## [Reviewer comments · BMJ Open]

ARTICLE DETAILS

TITLE (PROVISIONAL)	Cost-effectiveness of decellularised bone allograft compared with fresh-frozen bone allograft for acetabular impaction bone grafting during a revision hip arthroplasty in the UK
AUTHORS	Cowell, Kern; Statham, Patrick; Sagoo, Gurdeep; Chandler, James; Herbert, Anthony; Rooney, Paul; Wilcox, Ruth; Fermor, Hazel

VERSION 1 – REVIEW

REVIEWER	Lopatina, Elena University of Calgary
REVIEW RETURNED	03-Jan-2023

GENERAL COMMENTS	Thank you for the opportunity to review this manuscript. This work aimed to estimate whether the potential health benefit of decellularised bone allograft outweighs their increased cost. This is a well-done study, and a well-written manuscript. A few (rather minor) suggestions for the authors are as follows: 1. Suggest presenting average costs of RHA with decellularized allograft and fresh-frozen allograft in the introduction to put the costs discussed in methods and results into perspective (page 4, lines 20-22).2. For readers who might be unfamiliar with NHS, it would be helpful to briefly outline what costs are covered by NHS (e.g., are drug costs and aids and adaptations covered and relevant from the NHS perspective?). Also, is cost of physician time included in inpatient and outpatient costs?3. It is said that “The ‘post-revision’ health state has a cost of £54.19 per patient per year.” Was it assumed that the cost in the ‘post-revision’ state was the same after revision and re-revision? If so, do authors anticipate that this assumption could bias results one way or another? Also, Arden et al, which is referenced as the source for the cost estimate, presented post-surgery costs for good and poor outcomes separately. Did the authors used the cost of good outcome cases as an estimate? If so, this assumption and its potential implications should be discussed.4. Suggest using the 2022 version of the NICE guide (reference 12).5. It is generally preferred that cost-effectiveness estimates are derived from a probabilistic analysis. Authors should consider adding a probabilistic analysis. Otherwise, it would be useful to justify why only deterministic analysis was completed.
---

REVIEWER	Gagala, Jacek Medical University of Lublin
REVIEW RETURNED	11-Feb-2023

GENERAL COMMENTS	I read your manuscript with interest and appreciation. The paper is
---

	well written and it compares the cost effectiveness of a well-established method of treatment of hip revision surgery (impaction bone grafting with fresh-frozen allografts) with the method that does not have any clinical results (decellularized bone graft). Although the paper fulfills the aim and scope of the manuscript (health economics) it is rather a paper for the journal like Medical Hypothesis.
--	---

VERSION 1 – AUTHOR RESPONSE

Reviewer: 1

1. Suggest presenting average costs of RHA with decellularized allograft and fresh-frozen allograft in the introduction to put the costs discussed in methods and results into perspective (page 4, lines 20-22).

Added overall surgery costs to introduction, see paragraph 2 of Introduction section.

2. For readers who might be unfamiliar with NHS, it would be helpful to briefly outline what costs are covered by NHS (e.g., are drug costs and aids and adaptations covered and relevant from the NHS perspective?). Also, is cost of physician time included in inpatient and outpatient costs?

Included in paragraph 1 of the method Costs section.

3. It is said that “The ‘post-revision’ health state has a cost of £54.19 per patient per year.” Was it assumed that the cost in the ‘post-revision’ state was the same after revision and re-revision? If so, do authors anticipate that this assumption could bias results one way or another? Also, Arden et al, which is referenced as the source for the cost estimate, presented post-surgery costs for good and poor outcomes separately. Did the authors used the cost of good outcome cases as an estimate? If so, this assumption and its potential implications should be discussed.

There is limited data on re-revision, therefore it was decided to use the same cost value for the post-revision years as the re-revision years. Additionally, the Briggs model for revision hip arthroplasty from Arden et al. 2017 was followed and this does not split the outcomes into good and poor, just successful and re-revision.

4. Suggest using the 2022 version of the NICE guide (reference 12).

Checked for any relevant differences and changed reference to 2022 NICE guide.

5. It is generally preferred that cost-effectiveness estimates are derived from a probabilistic analysis. Authors should consider adding a probabilistic analysis. Otherwise, it would be useful to justify why only deterministic analysis was completed.

Added probabilistic analysis in the form of a stochastic model with scatter plot (Figure 2) and CEAC (Figure 3), indicating corroborating results to the base case model. Added Stochastic methods and results section and added discussion comments to paragraph 1 in Discussion section.

Reviewer: 2

Dear Authors,

I read your manuscript with interest and appreciation. The paper is well written and it compares the cost effectiveness of a well-established method of treatment of hip revision surgery (impaction bone grafting with fresh-frozen allografts) with the method that does not have any clinical results (decellularized bone graft). Although the paper fulfills the aim and scope of the manuscript (health economics) it is rather a paper for the journal like Medical Hypothesis.

BMJ open has an excellent history of publishing health economic models, and we feel this evaluation would be of interest to the readers of BMJ open.